# Alternating Gradient Descent and Mixture-of-Experts for Integrated Multimodal Perception

**Hassan Akbari**[*]   **Dan Kondratyuk**[*]   **Yin Cui**
**Rachel Hornung**   **Huisheng Wang**   **Hartwig Adam**

Google Research
{hassanak, dankondratyuk, yincui, rachelhornung, huishengw, hadam}@google.com

## Abstract

We present Integrated Multimodal Perception (IMP), a simple and scalable multimodal multi-task training and modeling approach. IMP integrates multimodal inputs including image, video, text, and audio into a single Transformer encoder with minimal modality-specific components. IMP makes use of a novel design that combines Alternating Gradient Descent (AGD) and Mixture-of-Experts (MoE) for efficient model & task scaling. We conduct extensive empirical studies and reveal the following key insights: 1) performing gradient descent updates by alternating on diverse modalities, loss functions, and tasks, with varying input resolutions, efficiently improves the model. 2) sparsification with MoE on a single modality-agnostic encoder substantially improves the performance, outperforming dense models that use modality-specific encoders or additional fusion layers and greatly mitigates the conflicts between modalities. IMP achieves competitive performance on a wide range of downstream tasks including video classification, image classification, image-text, and video-text retrieval. Most notably, we train a sparse IMP-MoE-L focusing on video tasks that achieves new state-of-the-art in zero-shot video classification: 77.0% on Kinetics-400, 76.8% on Kinetics-600, and 68.3% on Kinetics-700, improving the previous state-of-the-art by +5%, +6.7%, and +5.8%, respectively, while using only 15% of their total training computational cost.

## 1 Introduction

The human perception system is profoundly multimodal. We perceive the world through the integration of a vast array of sensory systems across domains — visual, auditory, olfactory, somatic, *etc*. Neurons for multimodal integration have been found in both multisensory convergence zones Calvert [2001] and unimodal regions Driver and Noesselt [2008] in the human brain. Studies in developmental psychology also suggest that interrelating simultaneous multimodal sensations is key for perceptual learning Smith and Gasser [2005]. Inspired by these findings, we see an opportunity for combined multisensory learning in machine learning systems as well.

The rapid rise of large-scale multitask frameworks and models [Raffel et al., 2020, Roberts et al., 2022, Radford et al., 2021, Yu et al., 2022, Wang et al., 2022] provides foundations for integrating capabilities that unify many disparate tasks under one model. However, given the vast quantity of independent variables involved in designing such a system, achieving an integrated multimodal machine learning model still remains an open research direction. More specifically, designing a multi-task model that integrates many multimodal signals is challenging due to various reasons: i. Different modalities require structurally different I/O signatures to properly train. ii. When training across multiple datasets, some modalities or objectives may not exist or cannot be applied, depending

---

[*]Equal contribution.

37th Conference on Neural Information Processing Systems (NeurIPS 2023).

on the input data and the task to perform. iii. The presence of multiple input modalities calls for careful considerations on the architectural design and allocation of parameters to certain modalities, often requiring extensive hyperparameter tuning to find the best use of computational resources.

Intuitively, as we scale a model, it becomes increasingly expensive to redesign the architecture or search for a better training objective. The issue is exacerbated in multimodal multi-task modeling, where we need to consider the combination of input modalities or datasets, loss functions, and tasks at large scales. Therefore, we would like to find a training approach that can be scaled incrementally: for any new task or objective, regardless of its input shape or output loss, we should be able to add it to the existing pretraining without compromising the previous tasks.

We navigate this problem by exploring ways in which we could train one multimodal model such that it (1) leverages as many existing datasets as possible, (2) can train on any combination of tasks or loss functions, and (3) does not slow down with the addition of any new dataset, task, or loss function. By solving all of these points simultaneously, a multimodal model could scale with an increasingly diverse and rich set of training data without needing to redesign the training framework when new tasks get integrated.

We observe in our empirical results that the combination of diverse, heterogeneous tasks that have been previously established as strong objectives individually (*e.g.*, supervised classification and self-supervised contrastive learning) across multiple modalities are not only complementary, but can offer better convergence than training on individual tasks. By implementing Alternating Gradient Descent (AGD) and Mixture-of-Experts (MoE) via recently developed JAX primitives, we enable our model to use a fraction of the computational cost and memory required by similar large-scale perception models [Radford et al., 2021, Jia et al., 2021, Yu et al., 2022], despite the addition of multiple modalities which would normally require 2-8× compute at similar batch sizes.

Given this context, our contributions and findings are as follows:

1. We define an integrated modality-agnostic encoder model, and leverage a strong combination of image-text contrastive, video-text contrastive, video-audio contrastive, and image/video/audio classification losses during pretraining to create an **I**ntegrated **M**ultimodal **P**erception (IMP) model, as shown in Figure 1.

2. Contrasting the conventional approach of summing the losses of multiple objectives, we show that alternating between objectives results in a design that allows seamless integration of virtually any number of tasks and datasets without significant memory overhead and results in better downstream evaluations.

3. We show that optimization between multiple heterogeneous multimodal tasks is complementary and results in a higher quality model than trained on any individual task.

4. To train on large batches of video and audio modalities without reducing training efficiency or loss of accuracy, we design a dynamic mixture of various resolutions, sequence lengths, and batch sizes throughout pretraining, and alternate training on all input variations.

5. We enable our model with MoE, showing strong performance gains compared to a more conventional multi-tower contrastive model, even when appplying MoE to both towers.

6. We scale our resulting MoE-IMP model to 2B sparse parameters with similar compute to ViT-L (300M parameters), resulting in state-of-the-art performance on several large-scale multimodal video understanding datasets.

## 2 Related Work

The optimality of AGD optimization vs. averaging the losses (or gradients) has been explored in prior work [Jain et al., 2017, Pascal et al., 2021]. Alternating multimodal multi-task training with AGD has been explored in PolyViT [Likhosherstov et al., 2021], which analyzes different methods to combining heterogeneous task in a single model. The work reports similar findings to our own, that combining objectives can be mutually beneficial and alternating between datasets weighted by the number of examples provides one of the best methods for optimization. Our work extends this to a much more generic setup supporting virtually any combination of modalities, tasks, resolutions, etc.

The use of sparse MoE for multimodal modeling can be seen in recent works like LIMoE [Mustafa et al., 2022], which uses a single MoE encoder for image-text tasks, and VL-MoE [Shen et al., 2023],

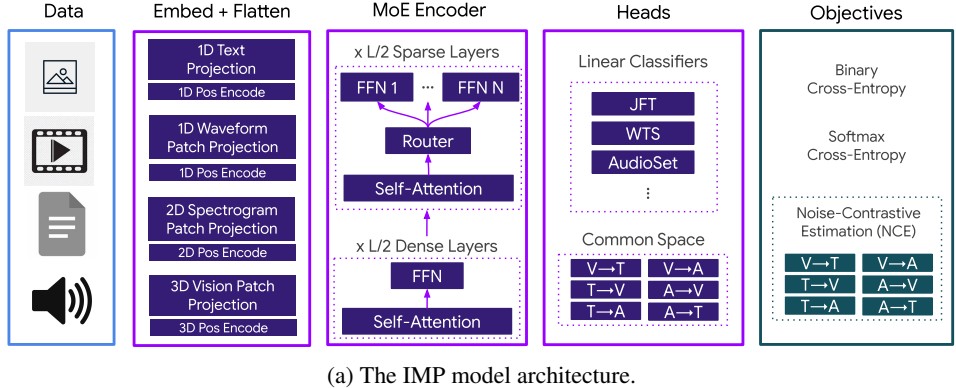

(a) The IMP model architecture.

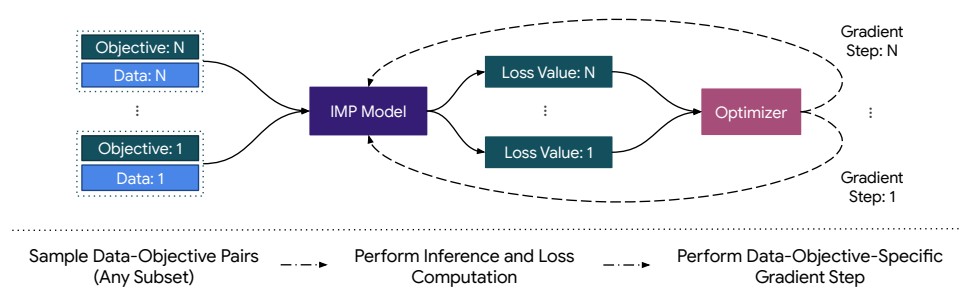

(b) The AGD-based multi-data multi-objective training overview.

Figure 1: **An overview of the IMP Training and Architecture.** A mixture of datasets with varying modalities, resolutions, and objectives are randomly sampled at each optimization step and fed into the model. The model is updated alternatingly given the data-objective pairs. We use `jax.jit` to compile and cache computation graphs to keep each step efficient while also allowing I/O shapes to change for every optimization step without requiring any costly padding or masking strategies.

which uses modality-specific experts for image-text modeling. Our work extends this concept further, introducing video and audio modalities with alternating training on multiple tasks and resolutions without requiring modality-specific experts.

Due to the inherent complexities of integrating modalities in one model, work has been done by simplifying the problem and focusing on a small set of universal objectives applicable to all modalities [Yu et al., 2022, Wang et al., 2022]. Alternatively, some works focused on applying padding or masking strategies to handle the incompatible I/O signatures from different tasks. We found in either case, this severely limits the ability for a model to leverage pre-existing large-scale datasets or to scale to entirely new modalities. The historical reasons are that: (i) most existing models are designed for specific input modalities *e.g.*, language [Brown et al., 2020, Chowdhery et al., 2022], vision [Dosovitskiy et al., 2020], or audio [Baevski et al., 2020]; (ii) different modalities are typically delegated to separate network weights for the best performance [Radford et al., 2021, Jia et al., 2021]; (iii) optimization difficulty with multiple modalities [Wu et al., 2022, Chen et al., 2022].

## 3  Method

### 3.1  Alternating Gradient Descent (AGD)

One of the core pillars of our approach to multimodal understanding is **task scalability**. I.e., different combinations of data and loss objectives should be interchangeable throughout training, while the addition of any new data or objective should not cause memory or computation overhead. We found that a common issue in training large-scale foundation models in a distributed setting is that input signatures and loss objectives need to be static to avoid major inefficiencies. Accelerated graph compilation APIs allow for low-level graph optimizations that maximize hardware FLOPs utilization on distributed devices such as GPUs and TPUs, but come at a cost of requiring static I/O signatures.

---

**Algorithm 1** Accelerated Multimodal AGD Algorithm

---

  **Input:** $M$ (model), $T$ (training steps), $X$ (dataset-objective pairs), $f$ (sampling function)
  **Initialize:** Model state $S_t$
  **while** $t \leq T$ **do**
    Sample data-objective pair $(D_t, L_t) \in X$ according to sampling function $f(t, S_t)$
    Compute forward pass predictions $P_t = jit(M(D_t))$
    Compute backwards pass on loss $jit(L_t(P_t, D_t))$
    Update model state $S_t$
  **end while**

---

One approach to handle the issue with the static input signature would be to use *mixed batching*, where all possible inputs are constructed, and inapplicable inputs for a given dataset are padded and outputs are masked accordingly at each training step. However, this comes at a great efficiency cost, since the more tasks that are added the more time is spent computing on padded inputs. The issue with having multiple objective functions is usually resolved by *mixed mini-batching*, where a batch is divided to multiple mini-batches with their corresponding objective functions. The gradients for each mini-batch and objective function pair are accumulated across multiple mini-batches and the model weights are updated once using an aggregated gradient. However, this approach is also difficult to scale since the gradients across multiple mini-batches are accumulated in memory and per-task batch size naturally reduces as we add more tasks.

We propose a more generic solution based on AGD [Jain et al., 2017], allowing any changes to the three elements of the optimization system: inputs, model, and objective. AGD can be seen as a superset of the conventional SGD, where at each gradient step, a different objective may be optimized given different sets of model weights and/or input modalities. More specifically, any input modality with arbitrary shape could consume any subset of the model while focusing on minimizing any specific combination of the objective functions. According to Jain et al. [2017] it is proven that if each of such optimization steps are convex individually, an alternation between them leads to a guaranteed convergence; as opposed to mini-batching where gradient accumulation could result in sub-optimal results. From a technical point of view, this approach requires compiling multiple computation graphs, one for each unique optimization step. To enable efficient execution of multiple computation graphs, JAX offers native Just-in-Time (JIT) compilation with the `jax.jit` API[2], which compiles the graph-of-interest at runtime and compiles a new graph if a change in structure is seen in any of the next optimization steps. Graphs themselves are cached by JAX in an in-memory lookup table so that tasks only need to be compiled once. We empirically observe that graph memory consumption constitute a negligible portion of the entire training. In our experiments, we tested up to 20 unique task structures with no significant reduction in training speed, with compilation taking only $0.34\%$ of the total training time, up to the largest scales.

The main AGD algorithm for our multimodal framework is provided in Algorithm 1. We note that we carefully design the loop to be as agnostic as possible with respect to the data-model-objective triplet. This allows for greater flexibility in defining logic inside the model to handle the processing of individual modalities and input shapes. The sampling function can also incorporate state from the optimization process itself. E.g., the loss value of a given step can be used as a reward signal to affect the sampling behavior in the next samplings [Piergiovanni et al., 2023, Mindermann et al., 2022]. In our default setup, we sample each unique task on a given step from a (single trial) Multinomial distribution with probabilities directly proportional to the number of examples in each task. We defer more complicated or reward-based settings to future studies.

### 3.1.1 AGD-Specific Efficiency Considerations

We notice that in each forward call, certain model states such as activations are stored by default to be used later in the backward call for gradient calculation. This is an established compute optimization at the low-level graph compilation in XLA and similar APIs. Although this trick helps a lot with the training time, it significantly consumes memory and creates memory overhead if more than one graph is compiled and used during training. To reduce memory overhead, we use JAX's

---

[2]`https://jax.readthedocs.io/en/latest/jax-101/02-jitting.html`

native **rematerialization** API, `jax.checkpoint`[3] to save memory by not checkpointing any of the intermediate model states. In our experiments, we observe an average reduction of 70-80% TPU HBM usage while resulting in only 18-20% longer step times.

We also notice that large models with many different objectives may still incur long compilation times. Therefore, we apply **scan-over-layers** with `jax.lax.scan`, a method which rolls all of the Transformer layers into a single layer that is called multiple times using different weights (instead of compiling the same function multiple times). This alone results in 15-30x faster compilation time depending on the model length. We observe increasing relative time savings with larger model sizes.

Furthermore, we accomplish these across distributed accelerators through the `jax.pjit` API[4], which distributes JIT compilation across multiple accelerators.

### 3.2 Objectives

Our goal in designing the IMP model is to reuse objectives that have been shown to be robust for learning each modality. Hence, we choose the two most established supervised and unsupervised learning objectives: i. Supervised Classification using Softmax/Binary Cross-Entropy(SCE/BCE), ii. Cross-Modal Noise-Contrastive Estimation (NCE). Unless otherwise specified, we do not sum any of the above losses or accumulate gradients as would be done traditionally. Instead we apply backprop on each objective individually with AGD.

### 3.3 Architecture

Figure 1 shows a high-level overview of the architecture of IMP, which consists of three main modules: i. The **Embedder**, which accepts specific modalities and embeds them in a shared modality-agnostic space; ii. The **MoE Encoder**, which computes semantic contextual embeddings from the embedded tokens; iii. The **Heads**, which produce all the final predictions from the Encoder by re-projecting its embeddings back into a modality-specific space. We briefly explain each module here and provide more details in Appendix.

One design decision important to multimodal modeling is how to allocate parameters to each modality. As seen in works like BASIC [Pham et al., 2021], an asymmetric modality-specific design can be more optimal than using a similar-sized model for each modality. However, this comes at the cost of requiring additional hyperparameter tuning to find the optimal parameterization. As we show later in the next section, we observe that through the use of model sparsification with MoE, a unified encoder design coupled with certain modality-specific pre- and post-encoder layers is more optimal than a traditional multi-encoder setup as seen in CLIP model variants.

We follow VATT [Akbari et al., 2021], AudioMAE [Huang et al., 2022], and T5 [Raffel et al., 2020] to extract the vision, audio, and text embeddings, respectively. We add learnable 3D/2D/1D positional encodings to the embeddings and project them to a space with the same dimensionality as the model's. We pass these embedded inputs regardless of modality as-is through the shared encoder, which is a standard Transformer architecture equipped with Mixture-of-Experts FFN layers. We follow V-MoE [Riquelme et al., 2021] and LIMoE [Mustafa et al., 2022] for expert allocation. This can be seen as an inductive bias, allowing each expert to be allocated to multiple modalities if the optimization benefits. One immediate benefit is that the addition of new modalities for fine-tuning does not need any specific changes to the encoder, unlike modality-specific experts which require additional modifications and input handling [Wang et al., 2022, Shen et al., 2023].

We apply modality-specific heads on the encoder representations to produce the final outputs for loss and prediction calculations. For classification objectives, we apply a dataset-specific linear classifier to the average-pooled outputs. For noise-contrastive estimation (NCE), we closely follow the CLIP architecture, applying separate feedforward heads for each modality-to-common-space projection.

### 3.4 Multi-Resolution Training

One major issue when training Transformers on video data is that computation and memory efficiency are usually bottlenecked due to Transformer's quadratic complexity as a function of the input length.

---

[3] `https://jax.readthedocs.io/en/latest/_autosummary/jax.checkpoint.html`

[4] `https://jax.readthedocs.io/en/latest/notebooks/Distributed_arrays_and_automatic_parallelization.html`

To counteract this, we propose to adjust batch size or resolution to compensate the additional temporal tokens, hence achieving a similar total number of input tokens compared to a single-frame still image.

We first fix a set tokens per batch $T = B \times T_F \times T_H \times T_W$, which is factorized by the batch size $B$, frame tokens $T_F$, height tokens $T_H$, and width tokens $T_W$ representing each patchified video. We observe that we can further factorize each batch by trading off different dimensions such that the total number of input tokens per step are roughly equal so that peak memory usage is preserved. For example, we can halve the spatial resolution while quadrupling the number of frames. This can increase convergence especially at the start of training, and provide a more memory efficient encoding of each objective. Furthermore, we leverage DropToken [Akbari et al., 2021] as an additional method to reduce tokens per batch by randomly dropping a fixed ratio of tokens per example. We find that for $T_F$ temporal frame tokens, we can randomly drop a ratio of $1 - \frac{1}{T_F}$ tokens per example to match the same tokens per batch as images. For certain objectives we find that a different mix of trade-offs is more optimal. For example, contrastive objectives favor large batch sizes, so we reduce the resolution or apply DropToken to be more memory efficient. On the other hand, classification objectives do not need as large batch sizes for optimal convergence, hence we reduce the batch size while increasing the spatiotemporal tokens.

## 4 Experiments and Results

### 4.1 Training Setup

**Datasets.** Our datasets consist of a diverse set of learnable signals across multiple modalities. We use WebLI [Chen et al., 2022], LAION-400M [Schuhmann et al., 2021], WIT [Srinivasan et al., 2021], CC12M [Changpinyo et al., 2021], and VCC [Nagrani et al., 2022] for vision-text contrastive learning; JFT-3B [Zhai et al., 2022], I21K [Ridnik et al., 2021], and WTS-70M [Stroud et al., 2020] for both supervised classification and label-based vision-text contrastive estimation (similar to BASIC [Pham et al., 2021]); HT100M [Miech et al., 2019] and AudioSet [Gemmeke et al., 2017] for vision-audio-text triplet contrastive loss (similar to VATT [Akbari et al., 2021]).

We use a proportionally weighted sampling algorithm, executing each task in succession. To ensure that datasets are evenly sampled, we weight each task by the number of examples, normalized to a probability distribution. For each dataset variant with different resolution sizes, we apply the same weight. For a fair evaluation on downstream tasks, we filter all near-domain examples from our pretraining datasets (about 5M examples total).

**Multi-Resolution Strategy.** In our experiments, we always configure the input parameters so that the number of frame tokens are always equal to 4. This will result in the base tokens per video batch being exactly 4x of image's. For video datasets, we construct three variants and uniformly sample from each variant during training: i. Reduce the resolution by half in each dimension, ii. Reduce the batch size by 4x, iii. Apply DropToken $d = 1 - \frac{1}{T_F} = 0.75$. For image datasets, we also apply a similar strategy but for the purpose of high-resolution learning. In addition to the base resolution, we have two extra variants: i. Reduce the batch size by 4x and double each spatial dimension, ii. Apply DropToken $d = 1 - \frac{1}{4} = 0.75$.

**Training Parameters.** For our final experiments, we train with a patch size of 4x16x16 on base input resolutions of 16x256x256 and 4x256x256 on video and image modalities respectively, resulting in a total of 1024 and 256 patches per sample. The text inputs in ImageNet21K and JFT are truncated to 16 tokens to improve step efficiency with no loss of information, while keeping the text length of the rest of the datasets to a maximum of 256 tokens. We use a base batch size of 65536 and train using the Adam optimizer, a peak learning rate of 1e-3 with a cosine schedule, and apply no weight decay. For MoE parameters, we apply experts-choose routing with a top-$c$ capacity factor of 1.0 and do not apply any jittering to the routing or other auxiliary losses. Training results in roughly 16B examples seen, or about 5T tokens. Taken together, these datasets represent about 11B unique image-text and video-audio-text examples.

During inference, we evaluate on the largest available resolution that the model was trained on, i.e., 16x512x512, and use a total of 8 clips per video at approximately 12.5 fps on all evaluated datasets.

| MODEL | PPT | TPU-DAYS | IN1K | C100 | K400 | K600 | K700 | UCF101 | HMDB51 | ESC-50 |
|---|---|---|---|---|---|---|---|---|---|---|
| CLIP [Radford et al., 2021] | 400M | - | 76.2 | - | - | - | - | - | - | - |
| CoCa-B [Yu et al., 2022] | 380M | 1.8k | 82.6 | - | - | - | - | - | - | - |
| X-CLIP [Ni et al., 2022] | 400M | - | - | - | 65.2 | - | - | 72.0 | - | - |
| BIKE [Wu et al., 2022] | 230M | - | - | - | - | 68.5 | - | 80.8 | 52.8 | - |
| Text4Vis [Wu et al., 2022] | 230M | - | - | - | 68.9 | - | - | 85.8 | - | - |
| AudioCLIP [Gowda et al., 2021] | 430M | - | - | - | - | - | - | - | - | **69.4** |
| **IMP-B** | 86M | 120 | 80.5 | 82.4 | 63.6 | 62.1 | 49.9 | 64.2 | 39.7 | 32.8 |
| **IMP-MoE-B** | 90M | 150 | 83.2 | 84.9 | 68.2 | 65.7 | 52.1 | 88.7 | 46.6 | 47.8 |
| **IMP-MoE-L** | 350M | 1.5k | **83.9** | **87.0** | **77.0** | **76.8** | **68.3** | **91.5** | **59.1** | 65.1 |
| **Large-scale models** | | | | | | | | | | |
| LIMoE [Mustafa et al., 2022] | 680M | - | 84.1 | - | - | - | - | - | - | - |
| LiT ViT-g [Chen et al., 2022] | 2B | - | 84.5 | 83.6 | - | - | - | - | - | - |
| CoCa [Yu et al., 2022] | 2B | 10k | 86.3 | - | - | - | - | - | - | - |
| VideoCoCa [Yan et al., 2022] | 2B | 10.5k | - | - | 72.0 | 70.1 | 62.5 | 86.6 | 58.6 | - |

Table 1: **Zero-Shot Classification** (top-1) results on image, video, and audio datasets. IMP achieves a new state-of-the-art on zero-shot video action recognition by a wide margin with significantly low training cost. Considering the total number of Parameters Per Token (PPT), IMP also significantly outperforms comparable models on zero-shot image classification.

## 4.2 Main Results

We scale up and tune IMP for best performance on video datasets and evaluate it on a variety of downstream tasks and datasets to understand how it generalizes to other modalities. Table 1 shows the zero-shot classification capabilities of the model on several image, video, and audio datasets.

We note that IMP significantly outperforms previous state-of-the-art regardless of the model size and sets new record on Kinetics [Kay et al., 2017, Carreira et al., 2018, 2019], UCF101 [Soomro et al., 2012], and HMDB-51 [Kuehne et al., 2011] top-1 accuracy. Compared to the previous state-of-the-art, VideoCoCa [Yan et al., 2022], we train IMP-MoE-L on 256 TPU v4 chips for 6 days, representing only 15% of the total training cost of VideoCoCa. Considering the total number of parameters per token (PPT), IMP also outperforms the previous comparable state-of-the-art model, CoCa, on ImageNet [Russakovsky et al., 2015] and CIFAR-100 with a relatively large margin. However, we observe that the model falls behind the state-of-the-art in zero-shot audio classification on ESC-50 [Piczak, 2015]. This might be explained by the fact that the total training examples for audio modality are almost negligible compared to image and video. Hence, the model has a very strong performance on video and image. We argue that this could be resolved by simply introducing more samples and a more balanced train scheduling method, which we differ to future studies.

## 4.3 Ablation

In this section, we hightlight experimentation with some key results which motivate the chosen set of features for our final IMP model. We refer the reader to Appendix for more ablation on several other aspects of the model. The experiments in this section use IMP-S or IMP-B trained for 250k steps with a base batch size of 8192. We set a fixed video/image resolution of 16x224x224/4x224x224 using a patch size of 4x16x16. Unless otherwise specified, we do not apply multi-scale resolution.

**Combined objectives are mutually beneficial.** We train the model on ImageNet21k with two objectives: Noise-Contrastive Estimation (NCE) and Softmax Cross-Entropy (SCE) and explore the following: i. train on the objectives separately, ii. combine the objectives by summing them, and iii. alternating (AGD) between the objectives on each step. In the case of alternating, for a fair comparison so that training time is equivalent, we fix the same number of steps (250k) so that each objective only optimizes 50% of the total steps (125k). We evaluate on ImageNet1k and CIFAR-100 by image-to-text retrieval and linear probing on the frozen model's features and report the results in Table 2. It is not a surprise that classification objective benefits fine-tuning evals the most, while contrastive objective benefits open vocabulary classification. However, we observe that combining both objectives is better than optimizing on them individually. And alternating between the objectives is better than non-AGD objective mixing. These results are similar to the findings of PolyViT [Likhosherstov et al., 2021], which report optimal performance on alternating the objectives, weighted by the size of each dataset. *This motivates us to fix one objective per training step and alternate optimization between them.*

| OBJECTIVE | IMAGENET1K | | CIFAR-100 | |
|---|---|---|---|---|
| | LINEAR | I → T | LINEAR | I → T |
| NCE | 39.8 | 46.7 | 84.0 | 49.8 |
| Softmax | 41.1 | - | 82.6 | - |
| NCE + Softmax, Sum | 47.6 | 46.7 | 82.4 | 51.6 |
| NCE + Softmax, Alternating | **49.9** | **48.0** | **84.1** | **52.4** |

Table 2: **Combining multiple objectives during pre-training.**. Alternating between both objectives offer the best performance, despite training on the same number of total steps.

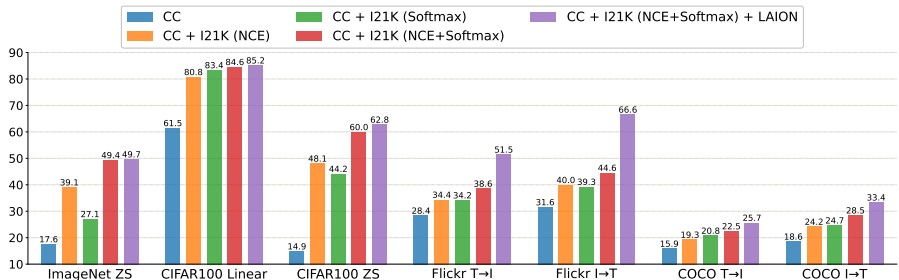

Figure 2: **Combining multiple datasets and objectives using AGD**. We integrate CC, I21k, and LAION with NCE and SCE using AGD and observe consistent improvement in downstream results. We also observe that NCE and SCE are mutually beneficial. Further optimality is provided by adding larger and more diverse datasets like LAION.

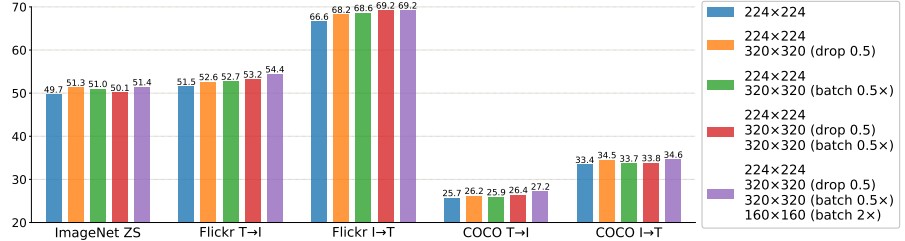

Figure 3: **Combining multiple input sizes using AGD**. We sample each variant with equal probability. Downstream results improve as we add more variants to the training mixture.

**Multi-task multi-dataset AGD is also mutually beneficial.** In Figure 2, we compare the result of adding additional datasets to the pretraining mixture. We additionally compare results across Flickr30k [Young et al., 2014] and COCO [Lin et al., 2014] datasets. We start with CC12M dataset and gradually add new datasets and objectives. Most notably, we compare the addition of I21K dataset, showing complementary improvement when combining NCE and SCE objectives. Similar to I21K isolated experiments, adding SCE benefits the entire pretraining mixture. While SCE benefits zero-shot results, NCE benefits linear probing results too. Certain dataset combinations (CC+I21K, CC+LAION) cause instability at the beginning of training. Adding a classification objective has a stabilizing effect, significantly reducing the chance of slow convergence. Optimizing on LAION directly is difficult, but benefits training a lot more when mixed in with other datasets. *This motivates us to further integrate a larger set of diverse datasets and objectives.*

**Multi-scale resolution provides universal improvement.** Figure 3 shows a comparison of using different combinations of resolution, batch size, and DropToken as input. In all settings, we fix the total tokens per batch, and we ensure that all training runs use the same number of total steps. We see that certain types of datasets respond well to DropToken while others may not. CC with double the batch size and DropToken 0.5 improves zero-shot image classification. Droptoken + 320x320 image on I21K SCE pretrain is better for linear probing and image-text retrieval. Adding multiple versions of smaller batch size + higher res, DropToken + higher res, larger batch size + lower res, can significantly improve downstream results. *We find that dynamic mixtures of resolution, batch size, and DropToken are always helpful.*

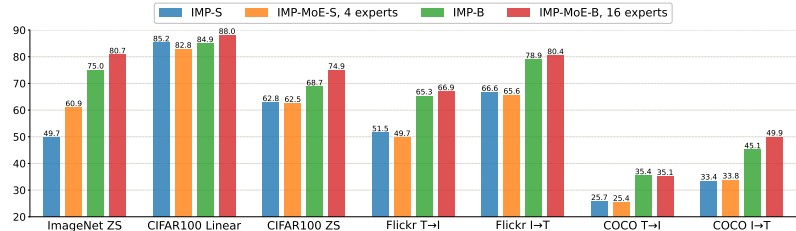

Figure 4: **IMP with Mixture-of-Experts**. Results show that using a modest 4 experts increases the model's accuracy substantially on ImageNet zero-shot evaluation. When we scale up the experts to 16, we see a consistent and significant improvement across all downstream evaluations.

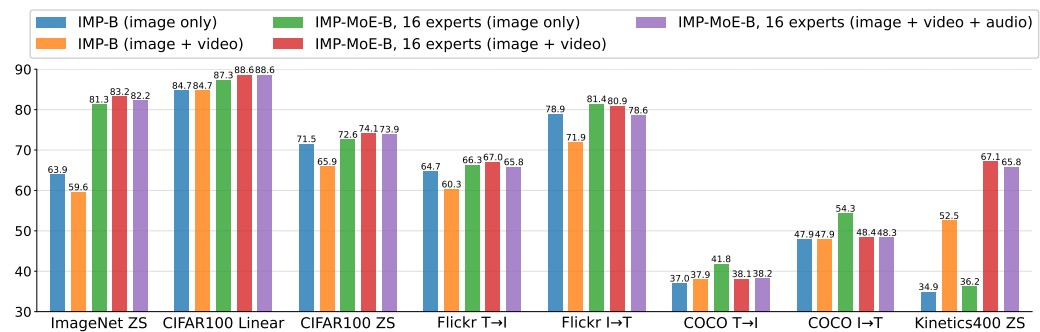

Figure 5: **Improved multimodal perception using MoE**. Results show significant improvement on diverse multimodal perception when we use the MoE variant of IMP. The addition of audio reduces accuracy on image and video metrics across the board, but is much less prominent when using MoE.

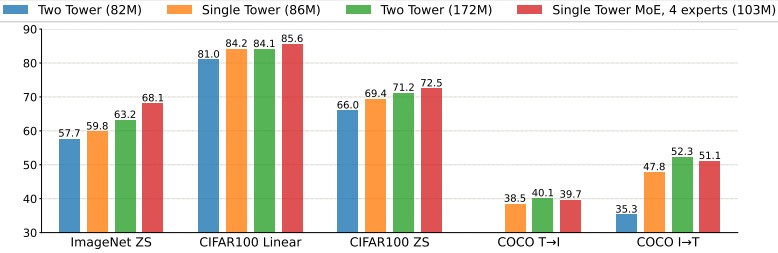

Figure 6: **Comparison of single-tower vs. multi-tower designs on IMP-B**. A single-tower MoE model is both parameter efficient and compute efficient compared to multi-tower dense variants.

**MoE provides universal improvement across modalities, and resolves the single-tower encoder parameter bottleneck.** The main challenge in designing a unified encoder tower as we have described is that parameters must be split between multiple modalities, thus harming accuracy. Compared to a two-tower contrastive model, the encoder of a unified image-text model contains half the parameters, while keeping training efficiency the same. One direction we explore is whether a large increase in parameters from MoE is sufficient to resolve parameter bottlenecks. In Figure 4, we observe that simply replacing a dense model with an equivalent MoE model with just 4 experts, we can provide a large gain in accuracy, especially for zero-shot metrics. This provides a promising indication that MoEs can be used to bridge the multimodal gap. We observe that with the addition of MoE, we can significantly close the gap between multiple modalities as seen in Figure 5. Since experts are free to choose which tokens are allocated to different experts, we observe strong alignment between experts and modalities.

**Single-tower MoE outperforms multi-tower dense variants.** We find that out of all the variants we tested, a unified MoE encoder provided the most parameter and compute efficient design, while significantly outperforming a multi-encoder modality-specific model in downstream results, as seen in Figure 6. When comparing two-tower models, we can either split the parameters to be roughly

equal in size to a single tower, or duplicate the towers to double the parameter count while providing equivalent computation. We observe that multi-tower dense models are more compute efficient than single-tower dense models, but less parameter efficient. However, a single-tower MoE model is both more compute and parameter efficient than all variants, showing improved generalization and using fewer parameters with the same compute budget as a multi-tower dense model. These results show universal superior parameter and compute efficiency and higher accuracy by using just 4 experts. This observation suggests that our method can be used for integrated multimodal multi-task modeling without worrying about the complications of modality-specific design choices or downstream performance degradation as observed in previous modality-agnostic designs [Akbari et al., 2021].

## 5 Conclusion

In this paper we presented an integrated training and modeling approach for multimodal perception using AGD and MoE. We observed that AGD enables task scalability and multi-resolution training, which improves the training convergence and the model's generalization capabilities. On the other hand, we found that MoE can play a very important role in integrating multiple modalities into one unified model. Given these findings, we scaled the model with hyperparamters tuned specifically for video understanding and achieved state-of-the-art performance in zero-shot video action recognition with a significant margin. Furthermore, we observed that the model also generalizes on other modalities and achieves competitive downstream results. In a nutshell, IMP opens a door to data (*e.g.*, modality, resolution, *etc.*) and task scalability — two important directions that have been neglected in many multimodal understanding works due to the inherent technical limitations. Due to the vast range of elements involved in this system, we defer multiple directions to be explored in future work: 1. generative objectives and model architectures, 2. causal MoE for generation, 3. sophisticated methods for data-objective sampling, 4. more downstream evaluations.

## Acknowledgments and Disclosure of Funding

We would like to thank Joan Puigcerver, Carlos Riquelme, and Basil Mustafa for their advice on MoE implementation and analysis; Anselm Levskaya for his help with advanced core JAX and Flax implementation; the T5X team for their support for scalable model partitioning, and Erica Moreira and Victor Gomez for their help with resource allocation.

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
