# Appendix: Alternating Gradient Descent and Mixture-of-Experts for Integrated Multimodal Perception

**Hassan Akbari**[*]  **Dan Kondratyuk**[*]  **Yin Cui**
**Rachel Hornung**    **Huisheng Wang**    **Hartwig Adam**

Google Research
{hassanak, dankondratyuk, yincui, rachelhornung, huishengw, hadam}@google.com

## A    Architecture

### A.1    Embeddings

For vision modalities, we use the VATT [Akbari et al., 2021] scheme to patchify each 3D video tensor. We define a video tensor of size $F \times H \times W$ with $F$ frames and $H \times W$ resolution using a patch size of $f \times h \times w$, producing $\frac{F}{f} \times \frac{H}{h} \times \frac{W}{w} \times 3$ voxels that are flattened into a single sequence. This is followed by linearly projecting the sequence into the model's hidden size. To allow for more robust generalization, we treat images as a special case of a video, assuming sequences are of shape $f \times H \times W$ and tiling frames $f$ times to fit in a single patch. For our base model, we use a patch kernel size of 4x16x16, up to 16 frames, and resolutions up to 512x512. Following VATT, to allow the model to adapt to different resolution scales, we apply learnable positional encodings to each patch position, which consist of the sum of 3 embedding tables along each separate axis: one for the temporal dimension, and two for the spatial dimensions of the video patches.

For text, we apply T5 encoding [Raffel et al., 2020] using the default English vocabulary with 32k SentencePiece tokens, which are embedded into the same hidden space size as image patch embeddings.

For audio, we apply both waveform and audio spectrogram as input. For spectrogram, following AudioMAE [Huang et al., 2022], after downsampling the audio waveform to 16000 kHz, we extract Mel-spectrograms with a duration of 8 seconds, producing 128 feature vectors with 128 dimensions each. We apply a patch kernel size of 16x16 to produce 64 total patches as input. For waveform, we use a kernel size of 256 samples and embed up to 256 tokens.

We use a separate learned positional embeddings for the linear sequence of text tokens and audio patches similar to the video encoding scheme above. To be able to handle different numbers of patches across different dimensions, the positional encoding of vision modalities needs to be handled with special care. Unlike the 1-dimensional sequences of text and audio waveform which can be truncated to a given length, the presence of 2D spatial dimensions mean that images with double the patches along a dimension should be subdivided into quadrants so that adjacent positions are close to each other in the embedding space. We accomplish this using a *dilated positional encoding*. For a given dimension a spatial positional encoding of $B$ buckets, if we encode a resolution with $P$ patches, we dilate the positional encoding with a stride of $\frac{B}{P}$. We treat spectrograms as 2D images and apply the same dilated encoding logic to them accordingly. In the case of the temporal dimension in video, we treat it the same as a 1-dimensional truncation independent of the spatial dimensions, which is applied in the same way for text.

---

[*]Equal contribution.

37th Conference on Neural Information Processing Systems (NeurIPS 2023).

| Model | Params (Dense) | Params (Sparse) | # Experts | # Layers | Hidden Size | FFN Size |
|---|---|---|---|---|---|---|
| **IMP-S** | 21M | 40M | 4 | 12 | 384 | 1536 |
| **IMP-B** | 86M | 400M | 16 | 12 | 768 | 3072 |
| **IMP-L** | 300M | 2B | 16 | 24 | 1024 | 4096 |

Table 1: **Comparison of IMP Architectures**. We provide parameters for dense and sparse MoE variants. Note that we only apply MoE to the last half of the layers in the encoder.

## A.2 MoE Encoder

For all MoE encoders, we use expert-choice routing [Zhou et al., 2022], which provides a strong baseline for all of the four modalities. Using expert-choice (top-$c$) routing, we observe much higher accuracy compared to the standard tokens-choose (top-$k$) routing. This is because experts-choose routing guarantees even load balancing, which we find to be an important factor for using an encoder shared across modalities. We find that only applying MoE to the last 50% of layers provided similar accuracy to applying them for all layers, therefore we use this setting for all MoE model variants.

We observe that contrastive optimization with MoE produces unstable output, often when introducing noisy text labels. This instability results in a loss divergence roughly within the first 30k-80k training steps. Similar to ViT-22B [Dehghani et al., 2023], we find that applying a layer normalization after the key and query matrices (QK LayerNorm) in the self-attention layers removes all such divergence issues in our training, hence we use this trick in all of our model variants.

## A.3 Heads

We apply global average pooling operation across the entire output sequence of the encoder, and use the resulting vector as the global features for classification and noise-contrastive estimation objectives. For classification objectives, we apply a dataset-specific linear classifier to the average-pooled outputs. For noise-contrastive estimation (NCE), we closely follow the CLIP architecture, applying separate feedforward heads for each modality-to-common-space projection. Each feedforward head consists of a two-layer linear projection with GeLU activation in between. The projection dimension size is the same as the model's hidden size.

## A.4 Model Sizes

Table 1 provides a description of model sizes. We provide results for three main variants, IMP-S, IMP-B, and IMP-L corresponding to encoder sizes of ViT-S, ViT-B, and ViT-L respectively [Zhai et al., 2022]. We also provide three additional sparse MoE sub-variants, which are indicated as IMP-MoE.

# B  Training Setup

## B.1  Datasets

For large-scale pretraining, we use the following datasets:

1. WebLI [Chen et al., 2022] consisting of 4B English-only image-text pairs. We use this dataset for image-text contrastive loss.

2. JFT-3B [Zhai et al., 2022], which contains a large collection of multi-class labels per image. We follow BASIC [Pham et al., 2021] for encoding multiclass indices as text and use the dataset for image-text contrastive loss as well as supervised classification loss.

3. LAION-400M [Schuhmann et al., 2021], a public dataset of 400M image-text pairs for image-text contrastive loss.

4. Wikipedia Image Text (WIT) [Srinivasan et al., 2021] with 37M image-text pairs sourced from Wikipedia for image-text contrastive loss.

5. Conceptual Captions (CC12M) [Changpinyo et al., 2021] consisting of 12 M image-caption pairs, used for image-text contrastive loss.

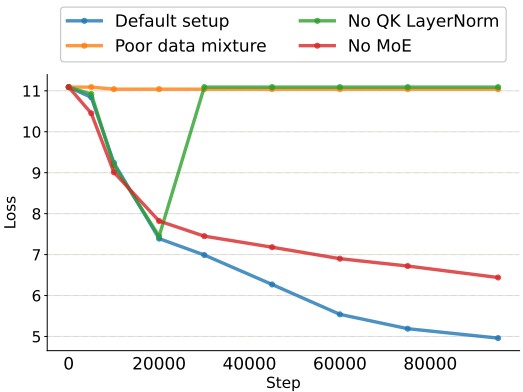

Figure 1: **Plot of loss with different training setups** in the first 100k steps of IMP-MoE-B. We observe that our MoE model with QK LayerNorm and using a diverse mixture of datasets reduces instability and produces the best loss convergence. A poor data mixture (e.g., CC + LAION) can cause a loss plateau, while adding QK LayerNorm in self-attention is important for avoiding loss divergence early in training.

6. ImageNet21K (I21K) [Ridnik et al., 2021] with 11M labeled images for image-text contrastive loss and supervised classification loss.

7. VideoCC (VCC) [Nagrani et al., 2022], a video dataset with a variant expanded to 1B English video-text pairs for video-audio-text triplet contrastive loss.

8. HowTo100M (HT100M) [Miech et al., 2019] consisting of ∼100M video-audio-ASR triplets, used for video-audio-text triplet contrastive loss.

9. Weak Text Supervision (WTS-70M) [Stroud et al., 2020], a dataset of 70M video clips obtained based on 700 action classes. We use this variant for video-text contrastive loss as well as supervised classification loss similar to JFT-3B and IN21k.

10. AudioSet [Gemmeke et al., 2017] for video-audio-text triplet contrastive loss.

## C   Additional Ablation

**Instabilities of contrastive loss on MoEs can be reduced with diverse data mixtures and QK LayerNorm.** We observe that a combination of MoE training with contrastive losses can lead to divergence, as seen in Figure 1. As seen in the figure (see also Table 4), the addition of multiple datasets, even under the same objective, can be detrimental to the optimization process. At the beginning of contrastive training on CC and LAION datasets, we observe a *loss plateau*, where the loss remains relatively constant from the start of training, and the model fails to start converging for a long period of time. On the other hand, if we apply the same dataset but add a softmax objective from the ImageNet21K dataset, we no longer observe a loss plateau, as softmax tends to be more stable for optimization processes than contrastive losses. This highlights the importance of selecting the right dataset mixture, especially at the start of training where the inherent nature of the random parameters can make it difficult for some task gradients to solidify a good direction in the optimization process.

We also observe certain divergences which occur early in training, and we found the magnitude of gradient updates can get large in the attention inputs. This can cause training to completely destabilize, and enter a similar loss plateau. Therefore, we apply QK LayerNorm, which we observe to have prevented such divergence across all of our experiments.

**Adding more modalities hurt single tower (dense) encoder accuracy.** We compare the addition of more modalities via video datasets in Figure 2. Adding a video dataset (i.e., WTS) to pretraining boosts Kinetics classification significantly, allowing the model to more easily discriminate between action classes. However, the addition of video data may harm image classification performance slightly, especially when parameters are constrained. Likewise, the addition of audio data with video has a slight negative impact, and the addition of dedicated audio classification dataset (i.e.,

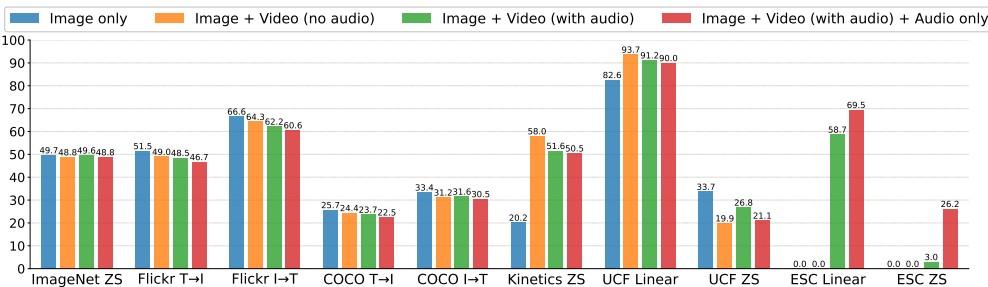

Figure 2: **Comparison of the addition of video and audio datasets on IMP-S**. The addition of video data (i.e., WTS) substantially improves video accuracy (Kinetics400, UCF) at the cost of slightly reducing the model's accuracy on image tasks (ImageNet, Flickr30k, COCO). The introduction of audio in the contrastive loss (i.e., AudioSet) also reduces both image and video accuracy slightly, but enables fine-tuning on audio data (ESC). Finally, the introduction of a dedicated audio class contrastive objective hurts image and video accuracy the most, but enables zero-shot audio classification.

| # TOWERS | # EXPERTS | ROUTER | PARAMS | IMAGENET ZS |
|:---:|:---:|:---:|:---:|:---:|
| 1 | 1 (dense) | N/A | 86M | 59.8 |
| 1 | 4 | tokens-choose | 103M | 62.5 |
| 2 | 4 | tokens-choose | 206M | 65.7 |
| 1 | 4 | experts-choose | **103M** | **68.1** |
| 2 | 4 | experts-choose | 206M | 68.4 |

Table 2: **Comparison of single tower MoE designs on IMP-B**, comparing experts-choose and tokens-choose approaches on ImageNet class retrieval. The most accurate and parameter efficient configuration is a single-tower experts-choose model. For tokens choose, we use a maximum capacity factor of 1.05 as in V-MoE so that training times are roughly equivalent.

AudioSet) has an even larger negative impact. This may be due to the additional audio-text contrastive signal which requires the network to allocate dedicated processing for audio-to-text understanding. *Therefore, a standard single tower encoder is not sufficient for optimal multimodal learning due to parameter bottlenecks.*

**Experts-choose routing is crucial for strong single tower performance.** We test the effect of experts-choose routing vs. tokens-choose in Table 2. Similar to findings in VL-MoE [Shen et al., 2023], we observe that separating experts by modality in the case of tokens-choose routing is useful for improving accuracy. However, when we switch to experts-choose routing, we find that performance increases further, and a multi-tower model is similar enough in accuracy that separating them per modality is no longer necessary. This allows for a much simpler model design, and we can fine-tune new modalities in the encoder without any additional setup or new experts required.

**Inserting diverse prompts during training helps improve zero-shot classification.** In Table 3, we evaluate different prompt settings during pretraining. Contrasting with prior works which only apply prompts during evaluation, we observe strong gains when randomizing the prompts during training. One interesting exception is in CIFAR-100, which benefits from training and testing on no prompts at all. We observe the trend that simpler prompts are more useful for datasets with a smaller number of classes. We typically care more about results of large-scale datasets, so we apply prompt diversification by default.

**Stability of optimization.** In general, we observe the following situations where optimization can become more unstable: (1) Increase in dataset diversity; (2) Increase in model size; (3) Increase in batch size. In Table 4, we show that training on NCE alone can cause instability issues during training, especially for noisy text datasets. But with the addition of more clean data sources and softmax objectives, we can greatly reduce the instability.

| CONFIGURATION | IN LINEAR | IN ZS | C100 LINEAR | C100 ZS | F30K T→I | F30K I→T | UCF LINEAR |
|---|---|---|---|---|---|---|---|
| No Prompt | 46.6 | 51.4 | **83.9** | **67.5** | 50.3 | **66.8** | 83.0 |
| CLIP Prompt | **47.7** | 55.8 | 83.7 | 57.8 | 49.7 | 66.5 | 81.9 |
| IMP Prompt | **47.7** | **56.5** | 83.8 | 60.5 | **51.2** | 66.3 | **84.1** |

Table 3: **Prompt comparison of IMP**. We compare three settings of train-time prompts with increasing diversity. Results show that randomized prompts in training tend to significantly improve metrics on classification tasks.

| DATASETS | IN LINEAR | IN ZS | C100 LINEAR | C100 ZS | F30K T→I | F30K I→T | COCO T→I | COCO I→T | UCF LINEAR |
|---|---|---|---|---|---|---|---|---|---|
| CC | 15.2 | 17.6 | 61.5 | 14.9 | 28.4 | 31.6 | 15.9 | 18.6 | 73.3 |
| CC + LAION | Diverged | - | - | - | - | - | - | - | - |
| CC + I21K (NCE only) | 33.5 | 39.1 | 80.8 | 48.1 | 34.4 | 40.0 | 19.3 | 24.2 | 78.3 |
| CC + I21K (Softmax only) | 42.5 | 27.1 | 83.4 | 44.2 | 34.2 | 39.3 | 20.8 | 24.7 | 82.4 |
| CC + I21K (NCE+Softmax) | **49.0** | 49.4 | 84.6 | 60.0 | 38.6 | 44.6 | 22.5 | 28.5 | **82.9** |
| CC + I21K (NCE+Softmax) + LAION | 46.9 | **49.7** | **85.2** | **62.8** | **51.5** | **66.6** | 25.7 | 33.4 | 82.6 |

Table 4: **Comparison of datasets & objectives with AGD on IMP-S**. We integrate Conceptual Captions (CC) with contrastive (NCE) loss, and ImageNet21K (I21K) with NCE and softmax loss. The addition of both NCE and Softmax objectives from classification-based pretraining datasets are mutually beneficial with the retrieval-based pretraining datasets, observing best performance with the combination of both objectives. Further optimality is provided by adding larger, more diverse dataset like LAION-400M. However, we find that LAION causes optimization on contrastive objectives to become unstable, so softmax loss can greatly stabilize this noisier dataset.

# D  Framework Modules

To make IMP possible, we have developed a framework for AGD which we call MAX, abbreviated from **M**ulti-task Multi-modal training based on J**AX**. MAX provides an end-to-end framework for running arbitrary multimodal data on models efficiently. An overview of modules used in MAX is provided in Figure 3.

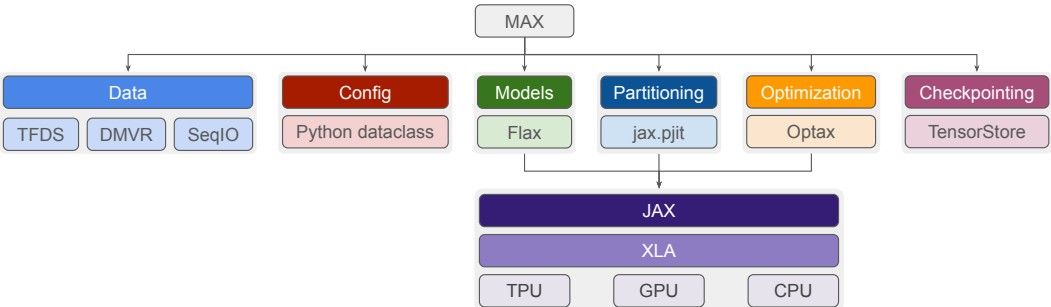

Figure 3: **Modules used to implement our MAX framework.**

The data pipeline defines data using TensorFlow Datasets (TFDS) [Abadi et al., 2015] and SeqIO [Roberts et al., 2022] registries for vision and language tasks. Preprocessing of text is provided by SeqIO, while image, video, and audio preprocessors are provided by DeepMind Video Readers (DMVR)[2]. Datasets are emitted from a `tf.data.Dataset` object provide a key-value signature that can be tightly integrated with models. This signature should be consistent with the model's expected input structure. For IMP, we define named keys for each modality and emit the applicable modalities from each dataset. Each modality can further provide optional metadata, information that specify how to properly execute or route the input to various modules.

---

[2]https://github.com/deepmind/dmvr

Models are built as native Flax[3] modules, partitioned `jax.pjit`[4], and optimized by transforms defined in Optax[5]. The JAX [Bradbury et al., 2018] framework provides a core selection of primitives that inferface with XLA[6], a library that compiles and optimizes computation graphs across different devices. We use TensorStore[7] to efficiently checkpoint and restore partitioned model parameters using async parallel dispatch. Configuration is specified according to Python dataclasses which can be overridden. This allows the creation of many variants of datasets, models, and experiments without excessive code duplication.

On each training step, the training loop samples a dataset-objective pair, passing inputs from the dataset directly into the model. Note that the routing of inputs across different model components is specifically avoided in the training loop logic to prevent the training process from being tied to a specific way to handle different input types. Instead, the model itself handles the interpretation of any combination inputs provided from the dataset and produces a named collection of outputs. Loss functions are applied in the training loop after sampling a dataset-objective pair and executing the model's forward pass. Together, this provides a modular way to interchange datasets, models, and loss functions.

We leverage training and inference step partitioning from `jax.pjit`, with further model and data parallelism abstractions provided by the t5x framework [Roberts et al., 2022] to partition model weights and activations across devices. On a high level, PJIT enables the use of dynamic graph compilation at runtime across many distributed devices. For each unique dataset-objective pair, PJIT will compile a new computation graph. These graphs are all cached so on subsequent iterations re-compilation overhead is minimized. This is used in conjunction with MoE to efficiently dispatch sparse weights across multiple devices while minimizing communication overhead. In conjunction with the partitioner, we initialize states by defining a set of specs of shapes that the model should accept as input, using `jax.eval_shape`. To efficiently run each training step, we pre-initialize the PRNG states of all training steps before any training takes place.

## E    IMP Model Card

We present the IMP model card in Table 5, following Mitchell et al. [2019].

| Model Summary | |
|---|---|
| Model Architecture | IMP is a multimodal sequence-to-sequence Transformer [Vaswani et al., 2017] encoder. It takes image, video, audio and text as inputs to the encoder and produces their feature embeddings as outputs. |
| Input(s) | RGB image, RGB video frame, audio waveform, audio spectrogram, text. |
| Output(s) | Feature embeddings corresponding to the inputs. |
| **Usage** | |
| Application | The model is for research prototype and the current version is not available for the broader public usage. |
| Known Caveats | No. |
| **System Type** | |
| System Description | This is a standalone model. |
| Upstream Dependencies | No. |
| Downstream Dependencies | No. |

---

[3]`https://github.com/google/flax`

[4]`https://jax.readthedocs.io/en/latest/jax.experimental.pjit.html`

[5]`https://github.com/deepmind/optax`

[6]`https://www.tensorflow.org/xla`

[7]`https://github.com/google/tensorstore`

| Implementation Frameworks | |
|---|---|
| Hardware & Software | Hardware: TPU [Jouppi et al., 2020]. |
| | Software: T5X [Roberts et al., 2022], JAX [Bradbury et al., 2018], Flaxformer[8], MAX |
| | Details are in Section D. |
| Compute Requirements | Reported in Section **??**. |
| **Model Characteristics** | |
| Model Initialization | The model is trained from scratch with random initialization. |
| Model Status | This is a static model trained on offline datasets. |
| Model Stats | The largest IMP model has 2B parameters for its sparse variant and 300M parameters for its dense variant. |
| **Data Overview** | |
| Training dataset | The model is pre-trained on the following mixture of datasets: Details are in Section **??**. |
| Evaluation and Fine-tuning Dataset | • **Image classification**: CIFAR, ImageNet
• **Video classification**: UCF101, HMDB51, Kinetics400, Kinetics600, Kinetics700
• **Audio classification**: ESC
• **Image to text / text to image retrieval**: Flickr30k, COCO |
| **Evaluation Results** | |
| Evaluation Results | Reported in Section **??**. |
| **Model Usage & Limitations** | |
| Sensitive Use | Reported in Section F |
| Known Limitations | Reported in Section F. |
| Ethical Considerations & Risks | Reported in Section F. |

Table 5: IMP model card.

# F   Limitations & Future Work

Our approach provides a promising new scaling direction that avoids many of the pitfalls when dealing with multimodal training. However, there are still some remaining obstacles from fully realizing this approach.

We note that our model provides exceptional performance in zero-shot video understanding, but falls slightly short in zero-shot image and audio understanding. We believe our training signals are have favored video understanding due to a combination of factors, including high incidence of vision data, a large sampling rate on vision tasks, and optimization losses converging faster on video. With a larger set of more diverse data and tasks (e.g., text-only and audio-only pretraining), we believe we can provide further improvements on these modalities without introducing any signficant training cost.

One unsolved question is how to best combine objectives during training. We have only tested configurations of tasks that are sampled equally across training. Instead, we can provide a more sophisticated curriculum to the model by scheduling tasks depending on the current step. There has

---

[8]https://github.com/google/flaxformer

been work showing further efficiency and accuracy improvements when scheduling different types of tasks at various stages [Wu et al., 2020, Piergiovanni et al., 2023].

Another obstacle is the use of multimodal MoE in the generative setting. Experts-choose routing has been integral to allowing a high performance single tower encoder model, but due to its requirement to aggregate tokens across the sequence, it is not by itself suitable for causal objectives like autoregressive sequence prediction as used in language modeling. Some additional modifications may make this possible, however.