# OpenReview forum: "Alternating Gradient Descent and Mixture-of-Experts for Integrated Multimodal Perception"
_NeurIPS.cc/2023/Conference — NeurIPS 2023 poster_

### Official Review · Reviewer_SjW7 · 2023-07-05

**Soundness:** 3 good
**Presentation:** 2 fair
**Contribution:** 2 fair
**Rating:** 5
**Confidence:** 3

**Summary:**

This work proposes a solution to train multimodal multi-task training and demonstrate the model trained via this strategy outperforms other approaches in zero-short learning problems. The proposed solution mainly relies on current solutions including AGD, JAX library, DropToken, etc. it seems that this work can be helpful in multimodal related works.

**Strengths:**

1) The proposed training strategy and solution can help related work in multimodal research area.
2) The training strategy demonstrated better performance on zero-short learning approach over other compared works.

**Weaknesses:**

1) If this work focuses on the training strategy, more studies on downstream tasks are needed.
2) Although the work focuses on the training solution, more analysis of multimodal experimental results analysis are needed. For example, the analysis of the improvement over other approaches,  especially what kind of cases the new approach help improve the performance?


**Questions:**

1) Could the author demonstrate the major difference compared to AGD? In line 119, the author mentioned that it’s a more generic solution based on AGD, however, not clear about the innovation and difference.
2) How can the proposed approach be applied on these cases where one or two modalities are missing?
3) Will the proposed approach handle the situation like training with several different datasets together? And how to deal with the difference  among them? Is there any other additional steps needed?

---

> ### Author Rebuttal · Authors · 2023-08-09
>
> Thanks for providing a review of our paper, here we try to answer some of the concerns and comments about the paper.
>
> >If this work focuses on the training strategy, more studies on downstream tasks are needed.
>
> What kind of downstream tasks are you looking for? We provide evaluation of zero-shot ImageNet, CIFAR, Kinetics, UCF, HMDB, and ESC in our main results table, which should already provide a large number of downstream benchmarks. Our ablations furthermore look at additional benchmarks in context of the various modalities. We also do not solely focus on the training strategy, but also highlight the importance of using the MoE architecture as a way to integrate many modalities into one model without harming performance, and that is confirmed in our results across many downstream evaluations and ablations.
>
> >Although the work focuses on the training solution, more analysis of multimodal experimental results analysis are needed. For example, the analysis of the improvement over other approaches, especially what kind of cases the new approach help improve the performance?
>
> We show specifically the improvement of our approach when comparing various aspects of the integration of different parts of our training strategy in various figures and tables (multimodal multitask AGD, Mixture-of-Experts, multi-scale resolution, etc.). We also compare with other similar models (for image-text MoE, we compare with LIMoE, for the previous SoTA on zero-shot video classification, we compare with VideoCoCa). See also our supplementary material for additional findings and results where we explain which methods are most important for our modeling and training strategies. If there's a more specific set of experiments or evaluations that is missing that weaken our claims, please report them so we can address them.`
>
> >Could the author demonstrate the major difference compared to AGD? In line 119, the author mentioned that it’s a more generic solution based on AGD, however, not clear about the innovation and difference.
>
> We propose a novel generalization of the approach which has not been explored in any other work. Due to technical limitations of previous implementations of AGD, we were unable to find other methods training efficiently on many modalities each with their own dynamic input shapes; the most popular techniques tend to waste computation on padding which we found to be a significant source of inefficiency (and graph compilation is quite important to be able to fully saturate the computation on an accelerator). By carefully constructing computation graphs that leverage primitives such as a compilation cache in a modality agnostic way, we can maximize effective training FLOPs utilization across distributed accelerators. Previous instantiations of AGD only considered a special case where all of the inputs are fixed in shape/size. We show that this feature is quite important especially to be able to train efficiently on many different modalities at the same time. This is a point we may need to emphasize more strongly, we can provide more results on the training efficiency when using our accelerated AGD vs. prior approaches.
>
> >How can the proposed approach be applied on these cases where one or two modalities are missing?
>
> This is the core part of our accelerated AGD technique. Because each step only optimizes on a single dataset and alternates training between them, we can define a dataset with only the modalities we want to specify for that step. For example, if we want to optimize on audio+video data, we can compile a computation graph on audio data as input and computes classification, another computation graph that has video data as input and also computes classification, and a third computation graph that computes audio-video contrastive loss. So long as there exists a dataset definition that matches the modalities/objectives of at least one of the computation graphs, then it can be used in training. So if we have any dataset with missing data/modalities, we just simply leverage the computation graph compatible with the losses where that modality is missing and sample that as a new task.
>
> >Will the proposed approach handle the situation like training with several different datasets together? And how to deal with the difference among them? Is there any other additional steps needed?
>
> Yes, this is precisely our approach, and the major reason why we can obtain state-of-the-art zero-shot results. Without being able to train multiple datasets with different subsets of modalities we would not be able to make our approach work. We explain in section 3 how defining datasets in separate compiled computation graphs can allow for efficient multitasking, only computing on the data that is available, without relying on any expensive padding or gradient masking that other works typically rely on. For a concrete example, we might start training step 0 by computing the loss for a video-text-audio contrastive task on VideoCC, followed by an image classification task on JFT on training step 1, and then an image-text contrastive task on CC12M on step 2. We can schedule these datasets/objectives in any order because the computation graphs for each are cached after they are encountered for the first time, this is all automatic due to the `jax.jit` implementation (and this should also work with `torch.compile`, but we have not implemented this).

---

> > ### Comment · Reviewer_SjW7 · 2023-08-19
> >
> > Thanks for the authors' response. I would like to keep my original rating.

---

### Official Review · Reviewer_SrPj · 2023-07-06

**Soundness:** 3 good
**Presentation:** 3 good
**Contribution:** 2 fair
**Rating:** 5
**Confidence:** 4

**Summary:**

This paper proposes Integrated Multimodal Perception, which can use image, video, and audio datasets for multimodal and multitask training. The method is scalable, benefiting from the alternating gradient descent as it can alternate diverse modalities, loss functions, and datasets to perform gradient descent. Moreover, using a mixture of experts can help handle different modalities with only a modality-agnostic encoder. The multi-resolution training accelerates the training by using different spatial resolutions, sequence lengths, and batch sizes.

**Strengths:**

1. This paper provides a generic multimodal training solution involving input, model, and objective and successfully trained models with a combination of image, video, and audio datasets.

2. It achieves SOTA zero-shot results on video action recognition.

3. The paper is well-written and easy to follow.

**Weaknesses:**

1. Ablation studies show that alternating between the objectives on each step is better than combining them by summing them. Is this true for all cases or only for large-scale pretraining? Many papers on various topics sum up multiple losses for training. For example, in semi-supervised learning, the loss general consists of two parts: supervised loss and unsupervised loss. We may need to give a proper context for this conclusion.

2. The method's two technical components, alternating gradient descent and mixture of experts, mainly follow previous works. This paper integrates them and applies them to a larger scale of pretraining.


**Questions:**

1. Algorithm 1 shows the sampling uses step t and model state S_t, but Lines 142-143 say that sampling is directly proportional to the number of examples in each dataset. That is, the more samples a task has, the more likely the dataset gets sampled. It would be better to make the algorithm consistent with the descriptions. Moreover, in Algorithm 1, should there be another mini-batch sampling given a chosen task? Should the sampling function f be only for task/dataset sampling?

2. Line 227 mentions filtering examples close to the downstream tasks. How is the filtering conducted?

3. Why the ablation study in Section 4.3 uses spatial resolution 224 while the main results in Table 1 use resolution 256 during training?

4. The paper mentions using JAX in several places, like jax.jit compiling graph-of-interest at runtime, jax.checkpoint saving memory by not checkpointing any of the 152 intermediate model states, and jax.lax.scan accelerating the graph compilation. I'd like to know whether Pytorch can also do so. I know Pytorch also has checkpointing functionality, and Pytorch 2.0 provides torch.compile() API.

5. In Figure 3, a 224x224 image can result in 196 tokens, while a 320x320 image corresponds to 400 tokens. After dropping 50% tokens, it has 200 tokens longers than 196 tokens. Do you use padding to make them have the same length in experiments?

6. Line 237 mentions a patch size of 4x16x16. What does the 4 represent? An image generally has 3 RGB channels. Shouldn't it be 3x16x16? Line 238 says image resolution is 4x256x256, which also confuses me.

**Limitations:**

The experiments seem not to show whether pretraining with one modality can help improve another modality's downstream performance. For example, whether the image pretraining datasets help improve the downstream video action recognition results and whether using the video pretraining datasets can boost the downstream performance on ImageNet-1K.

---

> ### Author Rebuttal · Authors · 2023-08-09
>
> Thank you for a very detailed review, we will try our best to address your comments.
>
> >Ablation studies show that alternating between the objectives on each step is better than combining them by summing them. Is this true for all cases or only for large-scale pretraining?
>
> This is a good point, and we are careful to note that this experiment is limited to the case of multimodal multitask pretraining, especially in the context of classification and contrastive losses. However, we should also note that we provide a theoretical basis in addition for our empirical results (see lines 124-127), i.e., it is proven that the gradients of tasks applied across multiple optimization steps are either equal or more marginally convex than summing the gradients, which can reduce the difficulty of the optimization problem. This is what motivated our initial experiment and works very well for our model setup, but we do not claim that alternating training would result in better downstream performance than summing gradients on any possible combination of tasks. We would argue that this is not a weakness, but simply another broad direction which is out of scope of the paper.
>
> >The method's two technical components, alternating gradient descent and mixture of experts, mainly follow previous works. This paper integrates them and applies them to a larger scale of pretraining.
>
> We propose a novel generalization of AGD which has not been explored in other works. Due to technical limitations of previous implementations of the approach, we were unable to find other methods training efficiently on many modalities each with their own dynamic input shapes; the most popular techniques tend to waste computation on padding which we found to be a significant source of inefficiency. By carefully constructing computation graphs that leverage primitives such as a compilation cache in a modality agnostic way, we can maximize effective training FLOPs utilization across distributed accelerators. Our accelerated AGD approach is also what unlocks the efficient use of multi-scale (multi-resolution) data, which to our knowledge, is a novel training approach applied to multimodal training not observed in other works.
>
> >Algorithm 1 shows the sampling uses step t and model state S_t, but Lines 142-143 say that sampling is directly proportional to the number of examples in each dataset. That is, the more samples a task has, the more likely the dataset gets sampled. It would be better to make the algorithm consistent with the descriptions.
>
> In this case, f(t, S_t) would simply compute over a probability distribution independent of t and S_t. We intentionally left this open to try with alternative sampling algorithms, such as [29], but we found the global sampling strategy to work well enough in combination with our other improvements. If you would prefer we reword this, we can do so.
>
> >Moreover, in Algorithm 1, should there be another mini-batch sampling given a chosen task? Should the sampling function f be only for task/dataset sampling?
>
> We consider the function f to only be for choosing the dataset-objective pair, the sampling function is operating on the current time step t and the optionally the previous model state. Once a task is chosen, then a minibatch is sampled from that dataset as is usually done.
>
> >Line 227 mentions filtering examples close to the downstream tasks. How is the filtering conducted?
>
> We use perceptual hashing as used in similar works (e.g., [46]) to make sure that duplicate images or frames are removed if they appear in the validation set to avoid data leakage. We can update with this information.
>
> >Why the ablation study in Section 4.3 uses spatial resolution 224 while the main results in Table 1 use resolution 256 during training?
>
> These ablations were mainly to try to closely match the resolutions and datasets as seen in literature so they can be more easily compared and reproduced. It is only our large-scale pretraining that we break from this kind of setup to try to achieve the best performance.
>
> >I'd like to know whether Pytorch can also do so.
>
> Yes, actually. We mostly speak to jax because our implementation is based on it. But analagously, it is possible to use `torch.compile` to construct an AGD training pipeline to achieve a similar effect, along with our `rematerialization` and `scan-over-layers` optimizations we mentioned.
>
> >Do you use padding to make them have the same length in experiments?
>
> No, we don't use padding. The sequence lengths are fairly close enough so we have roughly the same compute/memory usage to not be an issue. In practice these two inputs would be compiled separately before alternating training on them.
>
> >Line 237 mentions a patch size of 4x16x16. What does the 4 represent?
>
> The 4 represents the temporal axis, so 4 frames are patched together in the same way the pixels are. But this also requires all inputs to be multiples of 4 frames, so in the case of images we inflate/tile them to 4 frames. This means that images are basically treated as 4-frame videos with no motion, from a modeling perspective.
>
> >The experiments seem not to show whether pretraining with one modality can help improve another modality's downstream performance. For example, whether the image pretraining datasets help improve the downstream video action recognition results and whether using the video pretraining datasets can boost the downstream performance on ImageNet-1K.
>
> We have results in Figure 2 in the supplementary material which indicates that integrating multiple modalities like image and video into a dense model tends to harm the accuracy of the other modalities. However, with Figure 5 in the main paper, we see that MoEs reverse this trend and improve. Most notably, the addition of video data actually improves the zero-shot performance of the image datasets, which was not true of the dense model.

---

> > ### Comment · Reviewer_SrPj · 2023-08-21
> >
> > Thank the authors for answering my questions. I will keep my original score. The authors can further improve the paper's quality by incorporating the answers.
> > * For example, removing f(t, S_t) from algorithm 1 to align with Lines 142-143 can make the algorithm easier to understand. You may discuss the other possible sampling options in texts. Adding a mini-batch sampling within one task in Algorithm 1 can make the logic smoother.
> > * The authors can consider adding a section describing the method's limitations and the context for some conclusions e.g., alternating among objectives is better than summing them. According to Figure 5,  adding new modalities can bring performance drops for other modalities on some tasks, which Reviewer tAnb also notes. Although both the modality competing and distribution are universal problems, they are also worth discussing explicitly in the paper.

---

> > > ### Author Response · Authors · 2023-08-21
> > > **Official authors' response to reviewer SrPj**
> > >
> > > We really appreciate the reviewer's suggestions. We definitely address these concerns upon acceptance of the paper.
> > > We have already mentioned these limitations in the paper, but would elaborate on them based on the reviewers' valuable feedback.

---

### Official Review · Reviewer_tYu8 · 2023-07-14

**Soundness:** 3 good
**Presentation:** 3 good
**Contribution:** 3 good
**Rating:** 5
**Confidence:** 4

**Summary:**

This paper proposes a scalable multimodal multitasking approach. It combines alternating gradient descent and mixture-of-experts to train a unified model. The extensive experiments verify the effectiveness of the proposed method. By scaling up the model, this method sets up a new state-of-the-art in zero-shot video classification.

**Strengths:**

- The proposed method achieves excellent performance in zero-shot video classification. It further confirms improving data and task scalability in multimodal learning is promising.

- This paper is well written and easy to understand.


**Weaknesses:**

- Using alternating gradient descent for efficient multimodal multitasking learning is not new, which has been explored in PolyViT. It seems that the proposed method is an extension with more engineering improvement. Also, the success of MoE architecture has been validated on image-text tasks. The contribution of this paper is to incrementally extend this architecture to video and audio modalities.

- Compared with other methods, the proposed methods use more training data. Does the improvement mainly come from the scale of the training data? If PolyViT is pretrained on a similar scale of the data, does the proposed method still have advantages?

- typo. Line 264, "differ" -> "defer"


**Questions:**

- In Figure 5, the audio modality contributes less to the performance improvement on the image (or video)-text tasks. How about the importance of image (or video) to audio-related tasks?


**Limitations:**

Yes, the authors have adequately addressed the limitations.

---

> ### Author Rebuttal · Authors · 2023-08-09
>
> Thanks for providing a detailed review of our paper, here we try to address some of the comments of the paper.
>
> >Using alternating gradient descent for efficient multimodal multitasking learning is not new, which has been explored in PolyViT. It seems that the proposed method is an extension with more engineering improvement.
>
> While AGD itself has been used in PolyViT, we propose a novel generalization of the approach which has not been explored in any other work. Due to technical limitations of previous implementations of the approach, we were unable to find other methods training efficiently on many modalities each with their own dynamic input shapes; the most popular techniques tend to waste computation on padding which we found to be a significant source of inefficiency. By carefully constructing computation graphs that leverage primitives such as a compilation cache in a modality agnostic way, we can maximize effective training FLOPs utilization across distributed accelerators. The addition of this along with multi-scale representations of each modality further boosts accuracy and efficiency at the same time, as shown in table 1. We also experiment with both classification and contrastive objectives, where we might have any combination of optional paired or unpaired data, something that PolyViT does not try.
>
> AGD when coupled with cross-modal learning is something we found to be very important to our results, as we can see knowledge transfer from videos to images and vice versa. We have results in Figure 2 in the supplementary material which indicates that integrating multiple modalities like image and video into a dense model tends to harm the accuracy of the other modalities. However, with Figure 5 in the main paper, we see that MoEs reverse this trend and improve. Most notably, the addition of video data actually improves the zero-shot performance of the image datasets, which was not true of the dense model. These results show that a good diversity of tasks and modalities that are enabled by our more flexible version of AGD, in combination with expanding the model capacity, are important for multimodal learning. We would not have been able to demonstrate such capability without leveraging the dynamic graph compilation and dataset-objective pair task sampling as described.
>
> >the success of MoE architecture has been validated on image-text tasks
>
> While concurrent work such as image-text has been shown on MoEs, we note that the addition of video and audio data is significantly more challenging to show improvement across all four modalities, and has been the primary focus of the paper, i.e., to demonstrate which scaling techniques are the most useful to integrate these modalities into a single model.
>
> > Does the improvement mainly come from the scale of the training data?
>
> Not all of our results can be explained by scaling up our data. We note that CoCa-B achieves 82.6% on ImageNet1k zero-shot classification while IMP-B, a similarly sized encoder (ViT-B) is 80.5%. However, IMP-MoE-B trained on the same data closes the gap with  83.2%, showing the practical effects of using the MoE architecture. Similarly, our ablations show that alternating training on multi-scale data is useful (figure 3), and multiple objectives (table 2, figure 2) in tandem also help. We admit the addition of more data is certainly helpful, but it is only the combination of all of our approaches (accelerated AGD, multi-scale data, multi-objective, MoE) that we see improvement that is capable of surpassing the state-of-the-art results across multiple multimodal benchmarks. If we simply train a PolyViT architecture on more data, our improvements would be fairly marginal (in figure 2, addition of the larger scale LAION dataset helps some benchmarks but on average the combination of multiple objectives help more).
>
> > In Figure 5, the audio modality contributes less to the performance improvement on the image (or video)-text tasks. How about the importance of image (or video) to audio-related tasks?
>
> We found that one significant challenge in training is in the integration of the audio modality. Especially without our AGD data mixture and MoE improvements, the performance declines significantly due to the difference in how audio data is learned vs. other modalities (see figure 2 in our supplementary material). Despite this, figure 5 shows a minimal penalty of our model on a few tasks, with the benefit that the model now has zero-shot audio capability that previously did not exist.

---

> > ### Comment · Reviewer_tYu8 · 2023-08-21
> >
> > I appreciate the authors' response. The response well addressed my concerns. I would keep my original rating.

---

### Official Review · Reviewer_tAnb · 2023-07-17

**Soundness:** 2 fair
**Presentation:** 3 good
**Contribution:** 2 fair
**Rating:** 3
**Confidence:** 4

**Summary:**

The paper proposed Integrated Multimodal Perception (IMP) for multi-modal multi-task learning. IMP consists of a shared Mixture-of-Experts (MoE) Encoder as well as modality-specific embedding layers and heads re-projecting representations to modality-specific space. Optimizing towards both supervised classification losses and self-supervised contrastive learning NCE losses, the authors proposed to adopt Alternative Gradient Descent (AGD) during training to accommodate different inputs, outputs and learning objectives without bringing too much memory / computation overhead when incorporating more tasks. Furthermore, a multi-resolution strategy was also proposed to train IMP on large batches of video and audio modalities without losing efficiency or accuracy, where various resolutions, sequence lengths, and batch sizes are used dynamically. Experiments on a good amount of public datasets show that the proposed method can achieve better or competitive performance on various downstream zero-shot classification tasks except audio classification on ESC-50. Ablation studies support some of the design choices.

**Strengths:**

1. The paper introduced a possible way to integrate arbitrary number of modalities into one model. The AGD + unified encoder + MoE design seems an interesting solution without losing efficiency and accuracy when incorporating more modalities **in some situations**.

2. The proposed dynamic resolution strategy is somewhat novel and effective.

3. Experiments on considerably various public datasets demonstrate the effectiveness of the proposed method on zero-shot classification when comparing with previous state-of-the-arts

4. Ablation studies on important model designs were conducted to give a more comprehensive understanding of the methods

**Weaknesses:**

1. One critical problem of IMP is that it seems to suffer from performance drop when incorporating new modalities as per to Fig. 5, possibly due to the shared encoder. Although MoE can alleviate it to some extent in some cases, it is not always working. For example, in Fig. 5, for COCO Text->Image and Image->Text, and K400 zero-shot, IMP can achieve better or similar performance when new modalities of videos are introduced, while IMP-MoE-16Exp obtained worse accuracy. This defect somewhat goes against the major claim of this paper.

2. Another problem is that the model performance on different tasks subjects to change depending on training sample distribution, according to Tab. 1 and Line 261-264. While tuning the distribution of samples of different modalities can improve audio classification, it is not sure whether or not performance on other modality classification will drop, based on Weakness 1. It is also not quite sure how robust the proposed method is against any changes of sample distribution, which may hinder the incorporation of more datasets in the future, going against the major claim again.

**Questions:**

While the paper's major claim/contribution is a new method for multi-modal multi-task learning which can integrate any number of modalities/tasks with arbitrary inputs/outputs/objectives, Weaknesses 1 and 2 are critical and should be addressed before an accept decision can be made, in my opinion. Please address them accordingly.

---

> ### Author Rebuttal · Authors · 2023-08-09
>
> Thank you for your detailed review, we will try to address any concerns encountered in the paper.
>
> >One critical problem of IMP is that it seems to suffer from performance drop when incorporating new modalities as per to Fig. 5
>
> We would like to note that this problem of performance drop when integrating new modalities is fairly universal across any model, assuming a fixed computation or parameter budget. The model is forced to balance all modalities across the available computation/parameter budget, so it's no surprise that a model could decrease in average performance with so many diverse modalities competing against each other. Without the integration of our techniques such as MoE, multi-scale data, multi-objective training, and accelerated AGD, we would observe a much steeper decline across all of our benchmarks. For example, we show in Figure 2 of our supplementary material a fairly steady reduction in accuracy after each new modality is integrated, while with MoEs in Figure 5 in the main paper this distribution is remarkably different. In fact, the zero-shot image performance improves with the presence of video data that did not happen without MoEs. So while we may see, e.g., COCO results look lower when comparing image only vs. image+video, our main claim is that, *on average*, the combination of all of our methods provide significantly better improvement in integrating all of these modalities into one model than previous approaches (while also enabling cross-modal knowledge transfer on four separate modalities). These relative reductions in accuracy tend to apply on the small scale but become less prominent when scaling the MoE model as seen in Table 1 (see also section C the supplementary material). We observe fluctuations may occur on very specific benchmarks but the average shows a steady improvement.
>
> We also note that a two-tower or multi-tower model may incur additional memory consumption as the parameters are replicated for each modality (see supplementary material), so we can typically scale our model larger than existing approaches. In Figure 5, we compare our model against itself so it mainly serves as as a way to gauge the influence of various modalities, but because our method also incorporates various aspects of training, when we compare vs. other models, the intersection of all of our various improvements (multi-task multi-objective multi-resolution data, MoEs, etc.) are all very important for surpassing the existing state-of-the-art. We are comparing a highly general multimodal model (one that trains on 4 modalities at the same time) against other larger models more specialized to one or two modalities, but despite that, our methods consume fewer resources and scale across these modalities much more easily.
>
> >Another problem is that the model performance on different tasks subjects to change depending on training sample distribution, according to Tab. 1 and Line 261-264
>
> Again, performance differences across changes in sample distribution is another universal problem for multimodal models, we would see very similar types of concerns for other models as well. Our paper is not specifically about providing a comprehensive solution to this type of problem but rather finding a method that is better than other competing methods. Our models in table 1 are all trained on the same data, so this comparison is not subject to change in the sample distribution, and we see that the use of MoEs and increase in model size tend to bridge large gaps that might be caused by the data distribution despite using the same data for training. We do provide results in Figure 2 and Figure 5 (and Figure 2 in the supplementary material) to suggest that on especially image and video datasets, these modalities provide mutual improvement when integrated in conjunction with all of our techniques, so scaling one larger would help the other. Audio deserves special consideration, as this modality is sufficiently different from the other modalities that we've observed any scale of audio data could cause accuracy drops on the other modalities. This is one reason why video+text multimodal works typically avoid modeling audio, as it requires special handling like a separate audio network. Instead, we show a much simpler approach of using a combined MoE tower to mitigate this. Therefore, we would argue that the approach is not negatively impacted by the sample distribution in the same way as competing works, as AGD in conjunction with MoEs and multi-scale data provide a robust way to maximize performance on all modalities even in the presence of highly unbalanced audio to video/image data.

---

> > ### Comment · Reviewer_tAnb · 2023-08-20
> > **The common problems are what researchers need to solve**
> >
> > I appreciate the authors rebuttal, although it doesn't address any of my concerns.
> >
> > As the authors also agree, what are pointed out in my review are common problems of modern methods and therefore need to addressed properly (I'm not saying "solved") by papers from good conferences like NeurIPS.
> >
> > What the authors are doing in this paper is first downgrading the performance of the baseline model by using a shared encoder and then showing improvements by using existing techniques. Assuming these techniques really work, the authors should apply these techniques to a better baseline model and make some real contribution to the community.
> >
> > This downgrading and improving thing cannot convince me of the merits of the paper. Unless my concerns are well addressed, I will argue for reject.

---

> > > ### Author Response · Authors · 2023-08-21
> > > **Official authors' response to reviewer tAnb**
> > >
> > > We appreciate the reviewer's comments on our responses.
> > > We would like to emphasize that the ultimate goal of our paper is misunderstood. The goal of our paper is not to solve the mentioned problem (data sample distribution), which is universal across all models and approaches. We rather introduce a collection of novel techniques that significantly improves training and efficiency of multimodal multi-task models (with many objectives and modalities) compared to previous established literature.
> > >
> > > We would also like to mention that the reviewer's statement about downgrading the model is incorrect. We do provide extensive experiments that show sharing an encoder is only a part of the final performance, since our method actually outperforms a model with dedicated encoders (please see Figure 3 in the Supplementary Materials).
> > >
> > > We would like to emphasize that the title of a venue does not change the fact that a problem is universal. We would like to humbly argue that the major goal of papers in such venues is to 1. Expose the community to certain fundamental problems, 2. Provide theoretical and/or practical solutions for such problems; both which we have presented and addressed in our paper. In this paper, we elaborate that by scaling the number of modalities and objectives we hit a certain degradation of performance. We provide solutions based on MoE and AGD to resolve such issues and support those solutions by extensive experiments.

---

### Decision · Program_Chairs · 2023-09-21

**Decision:**

Accept (poster)

**Comment:**

The paper received mixed reviews with 3 borderline accepts and 1 reject. Given the borderline nature of the reviews, the AC reviewed the whole paper carefully and would rank it as 7. After further discussions with the SAC, the AC recommends the paper for acceptance. Most reviewers (including the AC) agreed that the AGD + unified encoder + MoE design provides an interesting solution to multi-modal training of modality-agnostic encoders. The paper also provides extensive experimental evaluation on a variety of different benchmarks, achieving impressive results, especially on zero-shot action recognition. Given the strong results obtained using an under-studied type of architecture, the AC and SAC agree that the paper should be accepted.

With that being said, the paper needs to be revised in the final version. One important concern raised by tAnb is the fact that when a single model is used (instead of MoE), sharing the model across modalities often hurts performance. While MoEs address this issue, the result still raises questions about the use of modality-agnostic encoders. A thorough discussion of these tradeoffs (modality agnosticity vs. model size vs. performance) would benefit the paper. One suggestion from the AC would be to use hybrid architectures with modality-specific stems followed by modality-agnostic modules. The authors could then assess the model with increasing levels of modality-agnostic computation.
Finally, please take into account other comments and suggestions for the camera-ready version. For example, consider moving the experiments supporting contribution 5 (in the intro) to the main manuscript (as opposed to supplementary material), as this is a critical finding of this work.